# IMPLICIT REGULARIZATION FOR GROUP SPARSITY

**Jiangyuan Li**[*]**, Thanh V. Nguyen, Chinmay Hegde**[†] **& Raymond K. W. Wong**[*]
[*]Texas A&M University
[†]New York University
`{jiangyuanli, raywong}@tamu.edu;`
`thanhng.cs@gmail.com; chinmay.h@nyu.edu`

## ABSTRACT

We study the implicit regularization of gradient descent towards structured sparsity via a novel neural reparameterization, which we call a *"diagonally grouped linear neural network"*. We show the following intriguing property of our reparameterization: gradient descent over the squared regression loss, without any explicit regularization, biases towards solutions with a group sparsity structure. In contrast to many existing works in understanding implicit regularization, we prove that our training trajectory cannot be simulated by mirror descent. We analyze the gradient dynamics of the corresponding regression problem in the general noise setting and obtain minimax-optimal error rates. Compared to existing bounds for implicit sparse regularization using diagonal linear networks, our analysis with the new reparameterization shows improved sample complexity. In the degenerate case of size-one groups, our approach gives rise to a new algorithm for sparse linear regression. Finally, we demonstrate the efficacy of our approach with several numerical experiments[1].

## 1 INTRODUCTION

**Motivation.** A salient feature of modern deep neural networks is that they are highly overparameterized with many more parameters than available training examples. Surprisingly, however, deep neural networks trained with gradient descent can generalize quite well in practice, even without explicit regularization. One hypothesis is that the dynamics of gradient descent-based training itself induce some form of implicit regularization, biasing toward solutions with low-complexity (Hardt et al., 2016; Neyshabur et al., 2017). Recent research in deep learning theory has validated the hypothesis of such implicit regularization effects. A large body of work, which we survey below, has considered certain (restricted) families of linear neural networks and established two types of implicit regularization — standard sparse regularization and $\ell_2$-norm regularization — depending on how gradient descent is initialized.

On the other hand, the role of *network architecture*, or the way the model is parameterized in implicit regularization, is less well-understood. Does there exist a parameterization that promotes implicit regularization of gradient descent towards richer structures beyond standard sparsity?

In this paper, we analyze a simple, prototypical hierarchical architecture for which gradient descent induces *group* sparse regularization. Our finding — that finer, *structured* biases can be induced via gradient dynamics — highlights the richness of co-designing neural networks along with optimization methods for producing more sophisticated regularization effects.

**Background.** Many recent theoretical efforts have revisited traditional, well-understood problems such as linear regression (Vaskevicius et al., 2019; Li et al., 2021; Zhao et al., 2019), matrix factorization (Gunasekar et al., 2018b; Li et al., 2018; Arora et al., 2019) and tensor decomposition (Ge et al., 2017; Wang et al., 2020), from the perspective of neural network training. For nonlinear models with squared error loss, Williams et al. (2019) and Jin & Montúfar (2020) study the implicit bias of gradient descent in wide depth-2 ReLU networks with input dimension 1. Other works (Gunasekar et al., 2018c; Soudry et al., 2018; Nacson et al., 2019) show that gradient descent biases the solution towards the max-margin (or minimum $\ell_2$-norm) solutions over separable data.

---

[1]Code is available on `https://github.com/jiangyuan2li/Implicit-Group-Sparsity`

| | NNs | Noise | Implicit vs. Explicit | Regularization |
|---|---|---|---|---|
| Vaskevicius et al. (2019) | DLNN | ✓ | Implicit (GD) | Sparsity |
| Dai et al. (2021) | LNN | ✗ | Explicit ($\ell_2$-penalty) | (Group) Quasi-norm |
| Jagadeesan et al. (2021) | LCNN | ✗ | Explicit ($\ell_2$-penalty) | Norm induced by SDP |
| Wu et al. (2020) | DLNN | ✗ | Implicit | $\ell_2$-norm |
| This paper | DGLNN | ✓ | Implicit (GD) | Structured sparsity |

Table 1: Comparisons to related work on implicit and explicit regularization. Here, GD stands for gradient descent, (D)LNN/CNN for (diagonal) linear/convolutional neural network, and DGLNN for diagonally grouped linear neural network.

Outside of implicit regularization, several other works study the inductive bias of network architectures under *explicit* $\ell_2$ regularization on model weights (Pilanci & Ergen, 2020; Sahiner et al., 2020). For multichannel linear convolutional networks, Jagadeesan et al. (2021) show that $\ell_2$-norm minimization of weights leads to a norm regularizer on predictors, where the norm is given by a semidefinite program (SDP). The representation cost in predictor space induced by explicit $\ell_2$ regularization on (various different versions of) linear neural networks is studied in Dai et al. (2021), which demonstrates several interesting (induced) regularizers on the linear predictors such as $\ell_p$ quasi-norms and group quasi-norms. However, these results are silent on the behavior of gradient descent-based training *without* explicit regularization. In light of the above results, we ask the following question:

> Beyond $\ell_2$-norm, sparsity and low-rankness, can gradient descent induce other forms of implicit regularization?

**Our contributions.** In this paper, we rigorously show that a *diagonally-grouped linear neural network* (see Figure 1b) trained by gradient descent with (proper/partial) weight normalization induces *group-sparse* regularization: a form of structured regularization that, to the best of our knowledge, has not been provably established in previous work.

One major approach to understanding implicit regularization of gradient descent is based on its equivalence to a mirror descent (on a different objective function) (e.g., Gunasekar et al., 2018a; Woodworth et al., 2020). However, we show that, for the diagonally-grouped linear network architecture, the gradient dynamics is beyond mirror descent. We then analyze the convergence of gradient flow with early stopping under orthogonal design with possibly noisy observations, and show that the obtained solution exhibits an implicit regularization effect towards structured (specifically, group) sparsity. In addition, we show that weight normalization can deal with instability related to the choices of learning rates and initialization. With weight normalization, we are able to obtain a similar implicit regularization result but in more general settings: orthogonal/non-orthogonal designs with possibly noisy observations. Also, the obtained solution can achieve minimax-optimal error rates.

Overall, compared to existing analysis of diagonal linear networks, our model design — that induces structured sparsity — exhibits provably improved sample complexity. In the degenerate case of size-one groups, our bounds coincide with previous results, and our approach can be interpreted as a new algorithm for sparse linear regression.

**Our techniques.** Our approach is built upon the *power reparameterization* trick, which has been shown to promote model sparsity (Schwarz et al., 2021). Raising the parameters of a linear model element-wisely to the $N$-th power ($N > 1$) results in that parameters of smaller magnitude receive smaller gradient updates, while parameters of larger magnitude receive larger updates. In essence, this leads to a "rich get richer" phenomenon in gradient-based training. In Gissin et al. (2019) and Berthier (2022), the authors analyze the gradient dynamics on a toy example, and call this "incremental learning". Concretely, for a linear predictor $\mathbf{w} \in \mathbb{R}^p$, if we re-parameterize the model as $\mathbf{w} = \mathbf{u}^{\circ N} - \mathbf{v}^{\circ N}$ (where $\mathbf{u}^{\circ N}$ means the $N$-th element-wise power of $\mathbf{u}$), then gradient descent will bias the training towards sparse solutions. This reparameterization is equivalent to a diagonal linear network, as shown in Figure 1a. This is further studied in Woodworth et al. (2020) for interpolating predictors, where they show that a small enough initialization induces $\ell_1$-norm regularization. For noisy settings, Vaskevicius et al. (2019) and Li et al. (2021) show that gradient descent converges to sparse models with early stopping. In the special case of sparse recovery from under-sampled

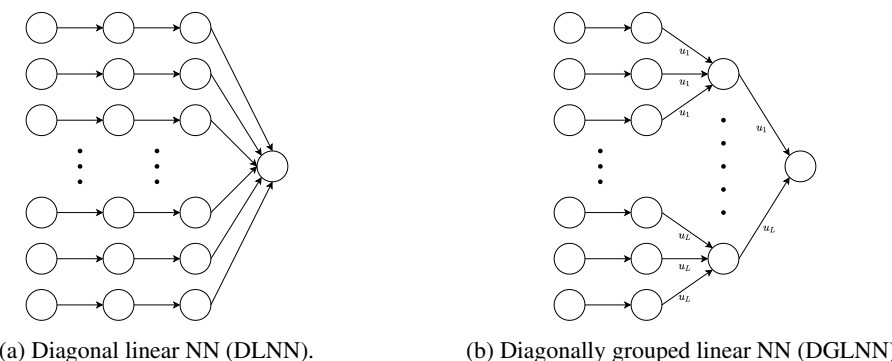

(a) Diagonal linear NN (DLNN).  (b) Diagonally grouped linear NN (DGLNN).

Figure 1: An illustration of the two architectures for standard and group sparse regularization.

observations (or compressive sensing), the optimal sample complexity can also be obtained via this reparameterization (Chou et al., 2021).

Inspired by this approach, we study a novel model reparameterization of the form $\mathbf{w} = [\mathbf{w}_1, \ldots, \mathbf{w}_L]$, where $\mathbf{w}_l = u_l^2 \mathbf{v}_l$ for each group $l \in \{1, \ldots, L\}$. (One way to interpret this model is to think of $u_l$ as the "magnitude" and $\mathbf{v}_l$ as the "direction" of the subvector corresponding to each group; see Section 2 for details.) This corresponds to a special type of linear neural network architecture, as shown in Figure 1b. A related architecture has also been recently studied in Dai et al. (2021), but there the authors have focused on the bias induced by an *explicit* $\ell_2$ regularization on the weights and have not investigated the effect of gradient dynamics.

The diagonally linear network parameterization of Woodworth et al. (2020); Li et al. (2021) does not suffer from identifiability issues. In contrast to that, in our setup the "magnitude" parameter $u_l$ of each group interacts with the norm of the "direction", $\|\mathbf{v}_l\|_2$, causing a fundamental problem of identifiability. By leveraging the layer balancing effect (Du et al., 2018) in DGLNN, we verify the group regularization effect implicit in gradient flow with early stopping. But gradient flow is idealized; for a more practical algorithm, we use a variant of gradient descent based on *weight normalization*, proposed in (Salimans & Kingma, 2016), and studied in more detail in (Wu et al., 2020). Weight normalization has been shown to be particularly helpful in stabilizing the effect of learning rates (Morwani & Ramaswamy, 2022; Van Laarhoven, 2017). With weight normalization, the learning effect is separated into magnitudes and directions. We derive the gradient dynamics on both magnitudes and directions with perturbations. Directions guide magnitude to grow, and as the magnitude grows, the directions get more accurate. Thereby, we are able to establish regularization effect implied by such gradient dynamics.

**A remark on grouped architectures.** Finally, we remark that grouping layers have been commonly used in grouped CNN and grouped attention mechanisms (Xie et al., 2017; Wu et al., 2021), which leads to parameter efficiency and better accuracy. Group sparsity is also useful for deep learning models in multi-omics data for survival prediction (Xie et al., 2019). We hope our analysis towards diagonally grouped linear NN could lead to more understanding of the inductive biases of grouping-style architectures.

## 2 SETUP

**Notation.** Denotes the set $\{1, 2, \ldots, L\}$ by $[L]$, and the vector $\ell_2$ norm by $\|\cdot\|$. We use $\mathbf{1}_p$ and $\mathbf{0}_p$ to denote $p$-dimensional vectors of all 1s and all 0s correspondingly. Also, $\odot$ represents the entry-wise multiplication whereas $\boldsymbol{\beta}^{\circ N}$ denotes element-wise power $N$ of a vector $\boldsymbol{\beta}$. We use $\mathbf{e}_i$ to denote the $i^{\text{th}}$ canonical vector. We write inequalities up to multiplicative constants using the notation $\lesssim$, whereby the constants do not depend on any problem parameter.

**Observation model.** Suppose that the index set $[p] = \cup_{j=l}^{L} G_l$ is partitioned into $L$ disjoint (i.e., non-overlapping) groups $G_1, G_2, \ldots, G_L$ where $G_i \cap G_j = \emptyset, \forall i \neq j$. The size of $G_l$ is denoted by $p_l = |G_l|$ for $l \in [L]$. Let $\mathbf{w}^\star \in \mathbb{R}^p$ be a $p$-dimensional vector where the entries of $\mathbf{w}^\star$ are non-zero only on a subset of groups. We posit a linear model of data where observations $(\mathbf{x}_i, y_i) \in \mathbb{R}^p \times \mathbb{R}$, $i \in$

$[n]$ are given such that $y_i = \langle \mathbf{x}_i, \mathbf{w}^\star \rangle + \xi_i$ for $i = 1, \ldots, n$, and $\boldsymbol{\xi} = [\xi_1, \ldots, \xi_n]^\top$ is a noise vector. Note that we do not impose any special restriction between $n$ (the number of observations) and $p$ (the dimension). We write the linear model in the following matrix-vector form: $\mathbf{y} = \mathbf{X}\mathbf{w}^\star + \boldsymbol{\xi}$, with the $n \times p$ design matrix $\mathbf{X} = [\mathbf{X}_1, \mathbf{X}_2, \ldots, \mathbf{X}_L]$, where $\mathbf{X}_l \in \mathbb{R}^{n \times p_l}$ represents the features from the $l^{\text{th}}$ group $G_l$, for $l \in [L]$. We make the following assumptions on $\mathbf{X}$:

**Assumption 1.** *The design matrix $\mathbf{X}$ satisfies*

$$\sup_{\|\boldsymbol{\beta}_1\| \leq 1, \|\boldsymbol{\beta}_2\| \leq 1} \left| \left\langle \boldsymbol{\beta}_1, \left( \frac{1}{n} \mathbf{X}_l^\top \mathbf{X}_l - \mathbf{I} \right) \boldsymbol{\beta}_2 \right\rangle \right| \leq \delta_{in}, \quad where \ \boldsymbol{\beta}_1, \boldsymbol{\beta}_2 \in \mathbb{R}^{p_l}, \tag{1}$$

$$\sup_{\|\boldsymbol{\beta}_1\| \leq 1, \|\boldsymbol{\beta}_2\| \leq 1} \left| \left\langle \frac{1}{\sqrt{n}} \mathbf{X}_l \boldsymbol{\beta}_1, \frac{1}{\sqrt{n}} \mathbf{X}_{l'} \boldsymbol{\beta}_2 \right\rangle \right| \leq \delta_{out}, \quad where \ \boldsymbol{\beta}_1 \in \mathbb{R}^{p_l}, \boldsymbol{\beta}_2 \in \mathbb{R}^{p_{l'}}, l \neq l', \tag{2}$$

*for some constants $\delta_{in}, \delta_{out} \in (0, 1)$.*

The first part (1) is a within-group eigenvalue condition while the second part (2) is a between-group block coherence assumption. There are multiple ways to construct a sensing matrix to fulfill these two conditions (Eldar & Bolcskei, 2009; Baraniuk et al., 2010). One of them is based on the fact that random Gaussian matrices satisfy such conditions with high probability (Stojnic et al., 2009).

**Reparameterization.** Our goal is to learn a parameter $\mathbf{w}$ from the data $\{(\mathbf{x}_i, y_i)\}_{i=1}^n$ with coefficients which obey group structure. Instead of imposing an explicit group-sparsity constraint on $\mathbf{w}$ (e.g., via weight penalization by group), we show that gradient descent on the *unconstrained* regression loss can still learn $\mathbf{w}^\star$, provided we design a special reparameterization. Define a mapping $g(\cdot) : [p] \to [L]$ from each index $i$ to its group $g(i)$. Each parameter is rewritten as $w_i = u_{g(i)}^2 v_i, \forall i \in [p]$. The parameterization $G(\cdot) : \mathbb{R}_+^L \times \mathbb{R}^p \to \mathbb{R}^p$ reads

$$[u_1, \ldots, u_L, v_1, v_2, \ldots, v_p] \to [u_1^2 v_1, u_1^2 v_2, \ldots, u_L^2 v_p].$$

This corresponds to the 2-layer neural network architecture displayed in Figure 1b, in which $\mathbf{W}_1 = \text{diag}(v_1, \ldots, v_p)$, and $\mathbf{W}_2$ is "diagonally" tied within each group:

$$\mathbf{W}_2 = \text{diag}(u_1, \ldots, u_1, u_2, \ldots, u_2, \ldots, u_L, \ldots, u_L).$$

**Gradient dynamics.** We learn $\mathbf{u}$ and $\mathbf{v}$ by minimizing the standard squared loss:

$$\mathcal{L}(\mathbf{u}, \mathbf{v}) = \frac{1}{2} \left\| \mathbf{y} - \mathbf{X}[(\mathbf{D}\mathbf{u})^{\circ 2} \odot \mathbf{v}] \right\|^2,$$

where

$$\mathbf{D} = \begin{pmatrix} \mathbf{1}_{p_1} & \mathbf{0}_{p_1} & \cdots & \mathbf{0}_{p_1} \\ \mathbf{0}_{p_2} & \mathbf{1}_{p_2} & \cdots & \mathbf{0}_{p_2} \\ \vdots & \vdots & \vdots & \vdots \\ \mathbf{0}_{p_L} & \mathbf{0}_{p_L} & \cdots & \mathbf{1}_{p_L} \end{pmatrix} \in \mathbb{R}^{p \times L}.$$

By simple algebra, the gradients with respect to $\mathbf{u}$ and $\mathbf{v}$ read as follows:

$$\nabla_{\mathbf{u}} L = 2\mathbf{D}^\top \left( \mathbf{v} \odot \left[ \mathbf{X}^\top \mathbf{X}((\mathbf{D}\mathbf{u})^{\circ 2} \odot \mathbf{v} - \mathbf{w}^\star) - \mathbf{X}^\top \boldsymbol{\xi} \right] \odot \mathbf{D}\mathbf{u} \right),$$

$$\nabla_{\mathbf{v}} L = \left[ \mathbf{X}^\top \mathbf{X}((\mathbf{D}\mathbf{u})^{\circ 2} \odot \mathbf{v} - \mathbf{w}^\star) - \mathbf{X}^\top \boldsymbol{\xi} \right] \odot (\mathbf{D}\mathbf{u})^{\circ 2}.$$

Denote $\mathbf{r}(t) = \mathbf{y} - \sum_{l'=1}^{L} u_l^2(t) \mathbf{X}_l \mathbf{v}_l(t)$. For each group $l \in [L]$, the gradient flow reads

$$\frac{\partial u_l(t)}{\partial t} = \frac{2}{n} u_l(t) \mathbf{v}_l^\top(t) \mathbf{X}_l^\top \mathbf{r}(t), \quad \frac{\partial \mathbf{v}_l(t)}{\partial t} = \frac{1}{n} u_l^2(t) \mathbf{X}_l^\top \mathbf{r}(t). \tag{3}$$

Although we are not able to transform the gradient dynamics back onto $\mathbf{w}(t)$ due to the overparameterization, the extra term $u_l(t)$ on group magnitude leads to "incremental learning" effect.

## 3 ANALYSIS OF GRADIENT FLOW

### 3.1 FIRST ATTEMPT: MIRROR FLOW

Existing results about implicit bias in overparameterized models are mostly based on recasting the training process from the parameter space $\{\mathbf{u}(t), \mathbf{v}(t)\}_{t \geq 0}$ to the predictor space $\{\mathbf{w}(t)\}_{t \geq 0}$ (Woodworth et al., 2020; Gunasekar et al., 2018a). If properly performed, the (induced) dynamics in the predictor space can now be analyzed by a classical algorithm: mirror descent (or mirror flow). Implicit regularization is demonstrated by showing that the limit point satisfies a KKT (Karush–Kuhn–Tucker) condition with respect to minimizing some regularizer $R(\cdot)$ among all possible solutions.

At first, we were unable to express the gradient dynamics in Eq. (3) in terms of $\mathbf{w}(t)$ (i.e., in the predictor space), due to complicated interactions between $\mathbf{u}$ and $\mathbf{v}$. This hints that the training trajectory induced by an overparameterized DGLNN may not be analyzed by mirror flow techniques. In fact, we prove a stronger negative result, and rigorously show that the corresponding dynamics *cannot* be recast as a mirror flow. Therefore, we conclude that our subsequent analysis techniques are necessary and do not follow as a corollary from existing approaches.

We first list two definitions from differential topology below.

**Definition 1.** *Let $M$ be a smooth submanifold of $\mathbb{R}^D$. Given two $C^1$ vector fields of $X, Y$ on $M$, we define the* Lie Bracket *of $X$ and $Y$ as $[X, Y](x) := \partial Y(x)X(x) - \partial X(x)Y(x)$.*

**Definition 2.** *Let $M$ be a smooth submanifold of $\mathbb{R}^D$. A $C^2$ parameterization $G : M \to \mathbb{R}^d$ is said to be commuting iff for any $i, j \in [d]$, the Lie Bracket $[\nabla G_i, \nabla G_j](x) = 0$ for all $x \in M$.*

The parameterization studied in most existing works on diagonal networks is separable, meaning that each parameter only affects one coordinate in the predictor space. In DGLNN, the parameterization is not separable, due to the shared parameter $\mathbf{u}$ within each group. We formally show that it is indeed not commuting.

**Lemma 1.** $G(\cdot)$ *is not a commuting parameterization.*

Non-commutativity of the parameterization implies that moving along $-\nabla G_i$ and then $-\nabla G_j$ is different with moving with $-\nabla G_j$ first and then $-\nabla G_i$. This causes extra difficulty in analyzing the gradient dynamics. Li et al. (2022) study the equivalence between gradient flow on reparameterized models and mirror flow, and show that a commuting parameterization is a sufficient condition for when a gradient flow with certain parameterization simulates a mirror flow. A complementary necessary condition is also established on the Lie algebra generated by the gradients of coordinate functions of $G$ with order higher than 2. We show that the parameterization $G(\cdot)$ violates this necessary condition.

**Theorem 1.** *There exists an initialization $[\mathbf{u}_{init}^\top, \mathbf{v}_{init}^\top] \in \mathbb{R}_+^L \times \mathbb{R}^p$ and a time-dependent loss $L_t$ such that gradient flow under $L_t \odot G$ starting from $[\mathbf{u}_{init}^\top, \mathbf{v}_{init}^\top]$ cannot be written as a mirror flow with respect to any Legendre function $R$ under the loss $L_t$.*

The detailed proof is deferred to the Appendix. Theorem 1 shows that the gradient dynamics implied in DGLNN cannot be emulated by mirror descent. Therefore, a different technique is needed to analyze the gradient dynamics and any associated implicit regularization effect.

### 3.2 LAYER BALANCING AND GRADIENT FLOW

Let us first introduce relevant quantities. Following our reparameterization, we rewrite the true parameters for each group $l$ as

$$\mathbf{w}_l^\star = (u_l^\star)^2 \mathbf{v}_l^\star, \quad \|\mathbf{v}_l^\star\|_2 = 1, \quad \mathbf{v}_l^\star \in \mathbb{R}^{p_l}.$$

The support is defined on the group level, where $S = \{l \in [L] : u_l^\star > 0\}$ and the support size is defined as $s = |S|$. We denote $u_{max}^\star = \max\{u_l^\star | l \in S\}$, and $u_{min}^\star = \min\{u_l^\star | l \in S\}$.

The gradient dynamics in our reparameterization does not preserve $\|\mathbf{v}_l(t)\|_2 = 1$, which causes difficulty to identify the magnitude of each $u_l$ and $\|\mathbf{v}_l(t)\|_2$. Du et al. (2018) and Arora et al. (2018)

show that the gradient flow of multi-layer homogeneous functions effectively enforces the differences between squared norms across different layers to remain invariant. Following the same idea, we discover a similar balancing effect in DGLNN between the parameter $\mathbf{u}$ and $\mathbf{v}$.

**Lemma 2.** *For any $l \in [L]$, we have*

$$\frac{d}{dt}\left(\frac{1}{2}u_l^2 - \|\mathbf{v}_l\|^2\right) = 0.$$

The balancing result eliminates the identifiability issue on the magnitudes. As the coordinates within one group affect each other, the direction which controls the growth rate of both $\mathbf{u}$ and $\mathbf{v}$ need to be determined as well.

**Lemma 3.** *If the initialization $\mathbf{v}_l(0)$ is proportional to $\frac{1}{n}\mathbf{X}_l^\top \mathbf{y}$, then*

$$\left\langle \frac{\mathbf{v}_l(0)}{\|\mathbf{v}_l(0)\|}, \mathbf{v}_l^\star \right\rangle \geq 1 - \left(\delta_{in} + L\delta_{out} + \left\|\frac{1}{n}\mathbf{X}_l^\top \boldsymbol{\xi}\right\|_2 / (u_l^\star)^2\right)^2.$$

Note that this initialization can be obtained by a single step of gradient descent with $\mathbf{0}$ initialization. Lemma 3 suggests the direction is close to the truth at the initialization. We can further normalize it to be $\|\mathbf{v}_l(0)\|_2^2 = \frac{1}{2}u_l^2(0)$ based on the balancing criterion. The magnitude equality, $\|\mathbf{v}_l(t)\|_2^2 = \frac{1}{2}u_l^2(t)$, is preserved by Lemma 2. However, ensuring the closeness of the direction throughout the gradient flow presents significant technical difficulties. That said, we are able to present a meaningful implicit regularization result of the gradient flow under orthogonal (and noisy) settings.

**Theorem 2.** *Fix $\epsilon > 0$. Consider the case where $\frac{1}{n}\mathbf{X}_l^\top \mathbf{X}_l = \mathbf{I}$, $\frac{1}{n}\mathbf{X}_l^\top \mathbf{X}_{l'} = \mathbf{O}, l \neq l'$, the initialization $u_l(0) = \theta < \frac{\epsilon}{2(u_{max}^\star)^2}$ and $\mathbf{v}_l(0) = \eta_l \frac{1}{n}\mathbf{X}_l^\top \mathbf{y}$ with $\|\mathbf{v}_l(0)\|_2^2 = \frac{1}{2}\theta^2, \forall l \in [L]$, there exists an lower bound and upper bound of the time $T_l < T_u$ in the gradient flow in Eq. (3), such that for any $T_l \leq t \leq T_u$ we have*

$$\left\|u_l^2(t)\mathbf{v}_l(t) - \mathbf{w}_l^\star\right\|_\infty \lesssim \begin{cases} \left\|\frac{1}{n}\mathbf{X}^\top \boldsymbol{\xi}\right\|_\infty \vee \epsilon, & \text{if } l \in S. \\ \theta^{3/2}, & \text{if } l \notin S. \end{cases}$$

Theorem 2 states the error bounds for the estimation of the *true* weights $\mathbf{w}^\star$. For entries outside the (true) support, the error is controlled by $\theta^{3/2}$. When $\theta$ is small, the algorithm keeps all non-supported entries to be close to zero through iterations while maintaining the guarantee for supported entries. Theorem 2 shows that under the assumption of orthogonal design, gradient flow with early stopping is able to obtain the solution with group sparsity.

## 4 GRADIENT DESCENT WITH WEIGHT NORMALIZATION

---
**Algorithm 1** Gradient descent with weight normalization

---
**Initialize:** $\mathbf{u}(0) = \alpha\mathbf{1}$, unit norm initialization $\mathbf{v}_l(0)$ for each $l \in [L]$, $\eta_{l,t} = \frac{1}{u_l^4(t)}$.

**for** $t = 0$ to $T$ **do**
    $\mathbf{z}(t+1) = \mathbf{v}(t) - \eta_{l,t}\nabla_\mathbf{v}\mathcal{L}(\mathbf{u}(t), \mathbf{v}(t))$
    $\mathbf{v}_l(t+1) = \frac{\mathbf{z}_l(t+1)}{\|\mathbf{z}_l(t+1)\|_2}, \forall l \in [L]$
    $\mathbf{u}(t+1) = \mathbf{u}(t) - \gamma\nabla_\mathbf{u}\mathcal{L}(\mathbf{u}(t), \mathbf{v}(t+1))$
    **if** the early stopping criterion is satisfied **then**
        stop
    **end if**
**end for**

---

We now seek a more practical algorithm with more general assumptions and requirements on initialization. To speed up the presentation, we will directly discuss the corresponding variant of (the more practical) gradient descent instead of gradient flow. When standard gradient descent is

applied on DGLNN, initialization for directions is very crucial; The algorithm may fail even with a very small initialization when the direction is not accurate, as shown in Appendix E. The balancing effect (Lemma 2) is sensitive to the step size, and errors may accumulate (Du et al., 2018).

Weight normalization as a commonly used training technique has been shown to be helpful in stabilizing the training process. The identifiability of the magnitude is naturally resolved by weight normalization on each $\mathbf{v}_l$. Moreover, weight normalization allows for a larger step size on $\mathbf{v}$, which makes the direction estimation at each step behave like that at the origin point. This removes the restrictive assumption of orthogonal design. With these intuitions in mind, we study the gradient descent algorithm with weight normalization on $\mathbf{v}$ summarized in Algorithm 1. One advantage of our algorithm is that it converges with *any* unit norm initialization $\mathbf{v}_l(0)$. The step size on $\mathbf{u}(t)$ is chosen to be small enough in order to enable the incremental learning, whereas the step size on $\mathbf{v}(t)$ is chosen as $\eta_{l,t} = \frac{1}{u_l^4(t)}$ as prescribed by our theoretical investigation. For convenience, we define $\zeta = 80 \left( \left\| \frac{1}{n} \mathbf{X}^\top \boldsymbol{\xi} \right\|_\infty \vee \epsilon \right)$, for a precision parameter $\epsilon > 0$. The convergence of Algorithm 1 is formalized as follows:

**Theorem 3.** *Fix $\epsilon > 0$. Consider Algorithm 1 with*

$$u_l(0) = \alpha < \frac{\epsilon^4 \wedge 1}{(u_{max}^\star)^8} \wedge \frac{1}{80L}(u_{min}^\star)^2 \wedge \frac{\epsilon}{L}, \quad \forall l \in [L],$$

*any unit-norm initialization on $\mathbf{v}_l$ for each $l \in [L]$ and $\gamma \leq \frac{1}{20(u_{max}^\star)^2}$. Suppose Assumption 1 is satisfied with $\delta_{in} \leq \frac{(u_{min}^\star)^2}{120(u_{max}^\star)^2}$ and $\delta_{out} \leq \frac{(u_{min}^\star)^2}{120s(u_{max}^\star)^2}$. There exist a lower bound on the number of iterations*

$$T_{lb} = \frac{\log \frac{(u_{max}^\star)^2}{2\alpha^2}}{2\log(1 + \frac{\gamma}{2}(\zeta \vee (u_{min}^\star)^2))} + \left\lfloor \log_2 \frac{(u_{max}^\star)^2}{\zeta} \right\rfloor \frac{5}{2\gamma(\zeta \vee (u_{min}^\star)^2)},$$

*and an upper bound*

$$T_{ub} \geq \frac{5}{16\gamma(\zeta \vee (u_{min}^\star)^2)} \log \frac{1}{\alpha^4},$$

*such that $T_{lb} \leq T_{ub}$ and for any $T_{lb} \leq t \leq T_{ub}$,*

$$\left\| u_l^2(t)\mathbf{v}_l(t) - \mathbf{w}_l^\star \right\|_\infty \lesssim \begin{cases} \left\| \frac{1}{n} \mathbf{X}^\top \boldsymbol{\xi} \right\|_\infty \vee \epsilon, & \text{if } l \in S \\ \alpha, & \text{if } l \notin S \end{cases}.$$

Similarly as Theorem 2, Theorem 3 states the error bounds for the estimation of the *true* weights $\mathbf{w}^\star$. When $\alpha$ is small, the algorithm keeps all non-supported entries to be close to zero through iterations while maintaining the guarantee for supported entries. Compared to the works on implicit (unstructured) sparse regularization (Vaskevicius et al., 2019; Chou et al., 2021), our assumption on the incoherence parameter $\delta_{out}$ scales with $1/s$, where $s$ is the number of non-zero groups, instead of the total number of non-zero entries. Therefore, the relaxed bound on $\delta_{out}$ implies an improved sample complexity, which is also observed experimentally in Figure 4. We now state a corollary in a common setting with independent random noise, where (asymptotic) recovery of $\mathbf{w}^\star$ is possible.

**Definition 3.** *A random variable $Y$ is $\sigma$-sub-Gaussian if for all $t \in \mathbb{R}$ there exists $\sigma > 0$ such that*

$$\mathbb{E}e^{tY} \leq e^{\sigma^2 t^2/2}.$$

**Corollary 1.** *Suppose the noise vector $\boldsymbol{\xi}$ has independent $\sigma^2$-sub-Gaussian entries and $\epsilon = 2\sqrt{\frac{\sigma^2 \log(2p)}{n}}$. Under the assumptions of Theorem 3, Algorithm 1 produces $\mathbf{w}(t) = (\mathbf{D}u(t))^{\circ 2} \odot \mathbf{v}(t)$ that satisfies $\|\mathbf{w}(t) - \mathbf{w}^\star\|_2^2 \lesssim (s\sigma^2 \log p)/n$ with probability at least $1 - 1/(8p^3)$ for any $t$ such that $T_{lb} \leq t \leq T_{ub}$.*

Note that the error bound we obtain is minimax-optimal. Despite these appealing properties of Algorithm 1, our theoretical results require a large step size on each $\mathbf{v}_l(t)$, which may cause instability at later stages of learning. We observe this instability numrerically (see Figure 6, Appendix E). Although the estimation error of $\mathbf{w}^\star$ remains small (which aligns with our theoretical result), individual entries in $\mathbf{v}$ may fluctuate considerably. Indeed, the large step size is mainly introduced to maintain a strong directional information extracted from the gradient of $\mathbf{v}_l(t)$ so as to stabilize the updates of $\mathbf{u}(t)$ at the early iterations. Therefore, we also propose Algorithm 2, a variant of Algorithm 1, where we decrease the step size after a certain number of iterations.

**Algorithm 2.** *Run Algorithm 1 with the same setup till each $u_l(t), l \in [L]$ gets roughly accurate, set $\eta_{l,t} = \eta$. Continue Algorithm 1 until early stopping criterion is satisfied.*

**Theorem 4.** *Under the assumptions of Theorem 3 with replacing the condition on $\delta$'s by $\delta_{in} \leq \frac{\sqrt{\zeta}(u_{min}^\star)^2}{120(u_{max}^\star)^3}$ and $\delta_{out} \leq \frac{\sqrt{\zeta}(u_{min}^\star)^2}{120s(u_{max}^\star)^3}$, we apply Algorithm 2 with $\eta_{l,t} = \frac{1}{u^4(t)}$ at the beginning, and $\eta_{l,t} = \eta \leq \frac{4}{9(u_{max}^\star)^2}$ after $\forall l \in [L], u_l^2(t) \geq \frac{1}{2}(u_l^\star)^2$, then with the same $T_{lb}$ and $T_{ub}$, we have that for any $T_{lb} \leq t \leq T_{ub}$,*

$$\left\| u_l^2(t)\mathbf{v}_l(t) - \mathbf{w}_l^\star \right\|_\infty \lesssim \begin{cases} \left\| \frac{1}{n}\mathbf{X}^\top \boldsymbol{\xi} \right\|_\infty \vee \epsilon, & \text{if } l \in S. \\ \alpha, & \text{if } l \notin S. \end{cases}$$

In Theorem 4, the criterion to decrease the step size is: $u_l^2(t) \geq \frac{1}{2}(u_l^\star)^2, \forall l \in [L]$. Once this criterion is satisfied, our proof indeed ensures that it would hold for at least up to the early stopping time $T_{ub}$ specified in the theorem. In practice, since $u_l^\star$'s are unknown, we can switch to a more practical criterion: $\max_{l \in [L]}\{|u_l(t+1) - u_l(t)|/|u_l(t) + \varepsilon|\} < \tau$ for some pre-specified tolerance $\tau > 0$ and small value $\varepsilon > 0$ as the criterion for changing the step size. The motivation of this criterion is further discussed in Appendix D. The error bound remains the same as Theorem 3. The change in step size requires a new way to study the gradient dynamics of directions with perturbations. With our proof technique, Theorem 4 requires a smaller bound on $\delta$'s (see Lemma 16 versus Lemma 8 in Appendix C for details). We believe it is a proof artifact and leave the improvement for future work.

**Connection to standard sparsity.** Consider the degenerate case where each group size is 1. Our reparameterization, together with the normalization step, can roughly be interpreted as $w_i \approx u_i^2 \, \mathrm{sgn}(v_i)$, which is different from the power-reparameterization $w_i = u_i^N - v_i^N, N \geq 2$ in Vaskevicius et al. (2019) and Li et al. (2021). This also shows why a large step size on $v_i$ is needed at the beginning. If the initialization on $v_i$ is incorrect, the sign of $v_i$ may not move with a small step size.

## 5 SIMULATION STUDIES

We conduct various experiments on simulated data to support our theory. Following the model in Section 2, we sample the entries of $\mathbf{X}$ i.i.d. using Rademacher random variables and the entries of the noise vector $\boldsymbol{\xi}$ i.i.d. under $N(0, \sigma^2)$. We set $\sigma = 0.5$ throughout the experiments.

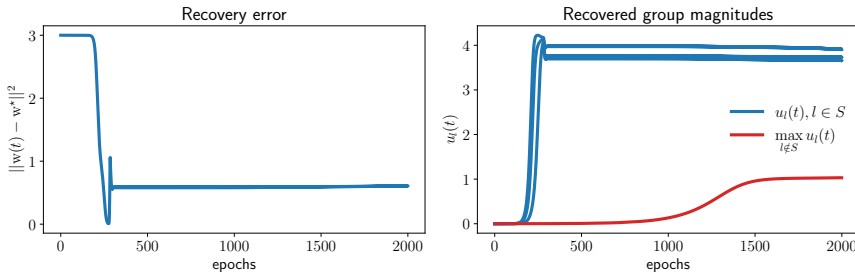

Figure 2: Convergence of Algorithm 1. The entries on the support are all 10.

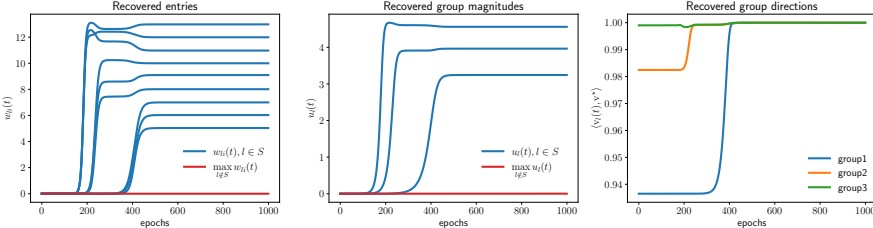

Figure 3: Convergence of Algorithm 2. The entries on the support are from 5 to 13.

**The effectiveness of our algorithms.** We start by demonstrating the convergence of the two proposed algorithms. In this experiment, we set $n = 150$ and $p = 300$. The number of non-zero entries is 9, divided into 3 groups of size 3. We run both Algorithms 1 and 2 with the same initialization $\alpha = 10^{-6}$. The step size $\gamma$ on $\mathbf{u}$ and decreased step size $\eta$ on $\mathbf{v}$ are both $10^{-3}$. In Figure 2, we present the recovery error of $\mathbf{w}^\star$ on the left, and recovered group magnitudes on the right. As we can see, early stopping is crucial for reaching the structured sparse solution. In Figure 3, we present the recovered entries, recovered group magnitudes and recovered directions for each group from left to right. In addition to convergence, we also observe an incremental learning effect.

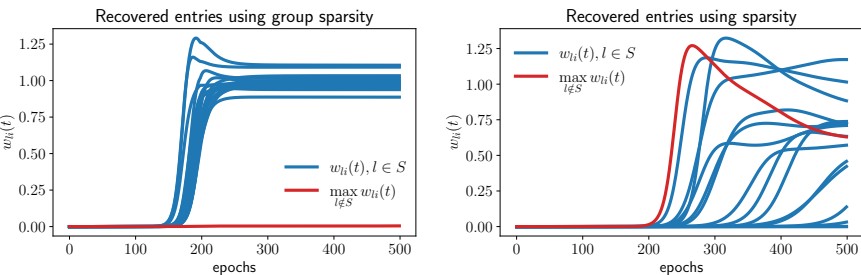

Figure 4: Comparison with reparameterization using standard sparsity. $n = 100, p = 500$.

**Structured sparsity versus standard sparsity.** From our theory, we see that the block incoherence parameter scales with the number of non-zero groups, as opposed to the number of non-zero entries. As such, we can expect an improved sample complexity over the estimators based on unstructured sparse regularization. We choose a larger support size of 16. The entries on the support are all 1 for simplicity. We apply our Algorithm 2 with group size 4. The result is shown in Figure 4 (left). We compare with the method in Vaskevicius et al. (2019) with parameterization $\mathbf{w} = \mathbf{u}^{\circ 2} - \mathbf{v}^{\circ 2}$, designed for unstructured sparsity. We display the result in the right figure, where interestingly, that algorithm fails to converge because of an insufficient number of samples.

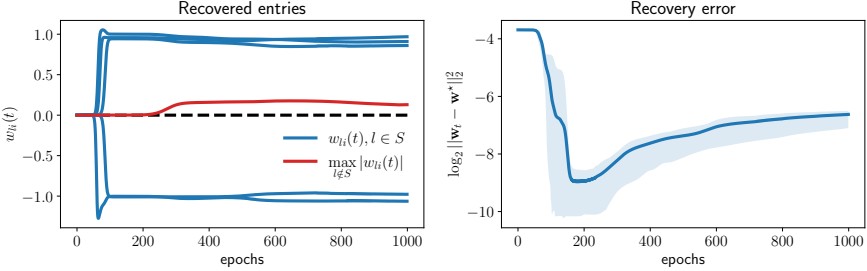

Figure 5: Degenerate case when each group size is 1. The $\log \ell_2$-error plot is repeated 30 times, and the mean is depicted. The shaded area indicates the region between the 25$^{\text{th}}$ and 75$^{\text{th}}$ percentiles.

**Degenerate case.** In the degenerate case where each group is of size 1, our reparameterization takes a simpler form $w_i \approx u_i^2 \text{sgn}(v)$, i.e., due to weight normalization, our method normalizes $v$ to 1 or $-1$ after each step. We demonstrate the efficacy of our algorithms even in the degenerate case. We set $n = 80$ and $p = 200$. The entries on the support are $[1, -1, 1, -1, 1]$ with both positive and negative entries. We present the coordinate plot and the recovery error in Figure 5.

## 6 DISCUSSION

In this paper, we show that implicit regularization for group-structured sparsity can be obtained by gradient descent (with weight normalization) for a certain, specially designed network architecture. Overall, we hope that such analysis further enhances our understanding of neural network training. Future work includes relaxing the assumptions on $\delta$'s in Theorem 2, and rigorous analysis of modern grouping architectures as well as power parametrizations.

ACKNOWLEDGMENTS

This work was supported in part by the National Science Foundation under grants CCF-1934904, CCF-1815101, and CCF-2005804.

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
