# OpenReview forum: "Implicit Regularization for Group Sparsity"
_ICLR.cc/2023/Conference — ICLR 2023 poster_

### Official Review · Reviewer_4y3z · 2022-10-21

**Confidence:** 3
**Correctness:** 4
**Technical Novelty And Significance:** 3
**Empirical Novelty And Significance:** 3
**Recommendation:** 8

**Clarity, Quality, Novelty And Reproducibility:**

Well-written and well-motivated paper.  Novelty is significant as the authors discover a particular neural net reparametrization that leads to group sparsity implicit regularization.  Results are reproducible as the code has been shared in the supplementary material.  I would kindly request the authors to make the code available in a public platform such as GitHub in case the paper gets accepted.

**Strength And Weaknesses:**

Strength
======

- Excellent contextualization of the problem being investigated.
- Thorough analysis of the proposed reparametrization and the reasons behind its implicit regularization structure.
- Great numerical analysis showcasing the theorectical development.
- The analysis in Theorem 1 showing that the gradient dynamics in DGLNN cannot be emulated by mirror descent is interesting and appreciated.

Weaknesses
=========

- No major weaknesses.

Minor Comments
=============

- If I were the authors I would be more specific in the title of the paper.  Maybe find a way to introduce "Neural Networks" in the title as it is the model being studied here?

**Summary Of The Paper:**

This paper studies "implicit regularization", a phenomena in gradient-based optimization algorithms in which the local minima of unregularized cost functions contain properties that one would expect only by adding a regularization term, e.g., sparsity being the most popular one. More precisely, they proposed a reparametrization in which the solution shows "group sparsity" structure, which is often obtained by having a elastic net regularization.

**Summary Of The Review:**

Well-organized paper with potentially impactful results.

---

> ### Author Response · Authors · 2022-11-15
> **Response to Reviewer 4y3z**
>
> Thank you for your encouraging review. We are very glad that you like this work.
>
> **Introduce neural networks in the title**
>
> Thank you for the suggestion. We considered it carefully but we would like to keep the title as it is; the corresponding NN architecture that we analyzed is relatively simple and mostly serves to improve our theoretical understanding of a novel implicit regularization effect. We do not want to overstate our contribution and give the impression that our theory is universally applicable to all network architectures.
>
> **Publish the code**
>
> Yes, we will make all our code public on GitHub after the review process.

---

> > ### Comment · Reviewer_4y3z · 2022-11-17
> > **Additional comments on title**
> >
> >  > We do not want to overstate our contribution and give the impression that our theory is universally applicable to all network architectures.
> >
> > That's is precisely why you would want to have a more accurate title. As it reads now, it's unclear that the implicit group sparsity behaviour is investigated in the context of diagonally grouped linear neural networks.

---

> > > ### Author Response · Authors · 2022-11-17
> > > **Response to additional comments on title**
> > >
> > > Thank you for the clarification. We agree that changing the title to ''Implicit Group Sparse Regularization in Diagonal Linear Networks'' would help to avoid confusion with the context. It’s not allowed to change the title with rebuttal revision. We are open to making the change after the review process.

---

> > > > ### Comment · Reviewer_aL9r · 2022-11-18
> > > > **point of view on the title**
> > > >
> > > > Although I appreciate reviewer's 4y3z enthusiasm, I find their review to be quite shallow and tend to stand with the authors on the "not overselling things" aspect. The method is new, yes, but as reviewer oXvD points out, the networks under scrutiny are not the ones that are popular in practice.
> > > > As the authors have pointed out this is classical in the literature, so not a reason for rejection, but I believe if you use neural networks in the title, you should also mention that they are of depth 2 to be honest.

---

### Official Review · Reviewer_aL9r · 2022-10-26

**Confidence:** 4
**Correctness:** 1
**Technical Novelty And Significance:** 3
**Empirical Novelty And Significance:** 2
**Recommendation:** 8

**Clarity, Quality, Novelty And Reproducibility:**

## Experiments:
- The authors use Rademacher random variables for X. It seems to me that using Gaussian entries is more standard. Random Gaussian matrices are also supposed to satisfy the Assumptions of the paper. For the rebuttal I'd like to see the same experiment with a random Gaussian matrix.
- "The entries on the support are all 1". Can the author provide the Signal to Noise Ratio, $||Xw^\star||/||\xi||$? That will be much more informative than saying $\sigma=0.5$.

## References:
- The following preprint would be relevant (it appeared on 31 August 2022, so it is of course not a flaw that it was not included in the paper): [1]

## Clarification.
- The authors state: "This reparameterization is equivalent to a diagonal linear network, as shown in Figure 1a." > For me Figure 1a use the parametrization $w = u \odot v$, that induces the same bias, but is not equal to the w= $u^N - v^N$ one. Where is $N$ in Figure 1a ? Where is the substraction?

## Maths:
- In section 3.2, $u^\star_{min}$ is potentially 0. Then in theorem 3, you need both $\delta$'s to be equal to 0?


[1] *Incremental Learning in Diagonal Linear Networks*, R. Berthier, https://arxiv.org/abs/2208.14673
[2] *Exact Regularization of Polyhedral Norms*, SIAMOPT 2010, F. Schopfer
[3] *Characterizing implicit bias in terms of optimization geometry*, ICML 2018, Gunasekar, Lee, Soudry, Srebro.
[4] *Iterative regularization for convex regularizers*, AISTATS 2021, C Molinari, M Massias, L Rosasco, S Villa

**Strength And Weaknesses:**

Strengths:
- this is a very active area of research, the contribution is relevant
- even though the assumptions are strong, there are not so different from classical ones in sparse recovery literature and are a first step to a better understanding of the phenomenon at stake

Weaknesses:
- the assumption is very strong and unlikely to hold in practice. Testing the behavior of the proposed algorithm on real data to assess the sensitivity to this assumption would be of great interest.
- comparison with explicit regularization methods is also expected. Implicit bias of mirror descent [2] for group elastic net could also be studied (if the elastic net regularization parameter is sufficiently low, the solution is the same as for group Lasso [2]), as well as iterative regularization methods based on primal dual procedures that can handle non strongly convex bias [4].

**Summary Of The Paper:**

The paper studies the implicit bias of gradient descent on diagonal linear networks.
It has been lately shown that parametrizing $x$ as $u \odot v$ in $y \approx A (u \odot v)$ induces sparsity in the gradient flow iterates on the least squares loss.
In the paper a special architecture is proposed so that gradient descent imposes **group sparsity** on the weights x$.
The architecture is a so-called *diagonally-grouped linear neural network* (DGLNN)
An analysis is provided under the restrictive assumption of orthogonal design (Theorem 2).
For $L$ groups, the proposed parametrization uses two set of variables: scalar ones $u_l$ encoding the magnitude, and vector ones $v_l$ encoding the direction.

Gradient flow of the overparametrized objective (Eq 3) is analyzed.
Section 4 is devoted to a practical version of the gradient flow, gradient descent with weight normalization.
Guaranties similar to the sparse recovery literature are derived under subGaussianity of the noise (Corollary 1)

**Summary Of The Review:**

Interesting idea and result, but valdiation fails to compare to the wealth of existing methods for sparse recovery, including implicit ones.

---

> ### Author Response · Authors · 2022-11-15
> **Response to Reviewer aL9r**
>
> We appreciate your thorough review of our paper. We will try to address your main concerns and questions here.
>
> **The assumption is very strong and unlikely to hold**
>
> Our early result (Theorem 2) requires an orthogonal design to analyze the gradient flow, which may not hold in practice. But we note that we have relaxed the condition to a within-group eigenvalue and block low-coherence condition when analyzing the gradient descent (Theorems 3 and 4). Note that we don’t require $\delta=0$. These coherence conditions are standard assumptions for analyzing group sparsity. That said, we agree that they are likely not the weakest. Our major focus is on gradient dynamics, which implies implicit regularization. We leave the relaxation for future work.
>
>
> **Comparison with explicit regularization methods, experiments using random Gaussian matrix and SNR**
>
> We have included more experiments using Gaussian random matrix, please see the updated Appendix E. The result doesn’t differ from experiments using the Rademacher random matrix. Moreover, we added comparisons with explicit regularization methods, proximal gradient descent, and the primal-dual procedure in the reference you mentioned. We use Gaussian design and vary the noise level to achieve different SNRs. The empirical results show that the proposed reparametrization method has a better performance. We agree that thorough experiments on real data would be an interesting future direction.
>
> **Figure clarification**
>
> $N$ represents the depth, so the figure is showing a specific choice of $N=3$. The figure is just an illustrative example of the parametrization $w=u^N$ assuming $w\geq0$. The subtraction is simply for handling both positive and negative entries, which does not affect the gradient dynamics.It is observed in practice and also shown theoretically that one part will always be small (ignorable). For example, when $w_j \geq0$, the corresponding $v_j$ is always very small, or when $w_j \leq 0$, $v_j$ is always small. In terms of understanding the implicit regularization effect (gradient dynamics), it doesn’t make any difference with or without subtraction.
>
> **$u_{min}^\star$ is potentially 0**
>
> This was a typo. The minimum is taken within the support on the group level. We have corrected it in the revision. We regret causing any confusion.
>
> **Missing reference**
>
> The reference you mentioned is closely related to the proof technique we use. We have added it in the revision. Thanks for pointing this out!

---

> > ### Author Response · Authors · 2022-11-25
> > **More results on real data**
> >
> > We conducted more experiments on real data.  We use the Bardet dataset provided in the R package "gglasso’" [1], which is originally from [2]. We randomly split the data equally, and use the validation dataset for hyperparameter tuning/early stopping. The result shown below demonstrates the efficacy of the overparameterized approach.
> >
> > |Test error|PGD|Primal-Dual|Our approach|
> > |---|:---:|:---:|:---:|
> > |**MSE**|0.03096 |0.02868|0.02477|
> >
> > Hope that we address all of your concerns. We look forward to your reply.
> >
> > [1] Yang and Zhou, A Fast Unified Algorithm for Computing Group-Lasso Penalized Learning Problems, Statistics and Computing, 2015.
> >
> > [2] Scheetz et al, Regulation of gene expression in the mammalian eye and its relevance to eye disease, PNAS, 2006.

---

> > > ### Comment · Reviewer_aL9r · 2022-12-12
> > > **Author's rebuttal**
> > >
> > > The authors rebuttal has addressed most of my concerns, I will therefore raise my grade.

---

### Official Review · Reviewer_oXvD · 2022-10-29

**Confidence:** 3
**Clarity, Quality, Novelty And Reproducibility:** Please see my comments above.
**Correctness:** 4
**Technical Novelty And Significance:** 2
**Empirical Novelty And Significance:** 3
**Recommendation:** 5

**Strength And Weaknesses:**

Strength:

1. This paper is well written and easy to read.

2. Some theoretical results in this paper are interesting.


Weaknesses:

1. As we know deep learning theory studies aim at understanding the modern neural networks in practice. The authors need to discuss the relationship between their results, such as the group sparsity structure,  and the observations of deep neural networks in practice.

2. In this paper, the authors focus on the diagonally grouped linear neural network, compared with the existing work, the authors need to explain more on the relationships between their structure and the ones in the modern neural networks. The grouped CNNs the authors mentioned are not the widely used neural network in practice. The authors are also recommended to specify the relationship between the structure of their model and grouped CNNs, such as where one can be reduced to another by using some specific settings.
The above limitation makes the contribution of this paper limited.

3. Can the authors show some experimental results to demonstrate that their findings also exist in the neural networks in real applications.



**Summary Of The Paper:**

In this paper, the authors study the theoretical properties of diagonally grouped linear neural networks, and they show that gradient descent over the squared regression loss, without any explicit regularization, biases towards solutions with a group sparsity structure. They conduct some experiments to verify their conclusion.  My main concern is the implications of these results on the modern neural networks are unclear. The reason is that the relationship between the diagonally grouped linear neural networks and the widely used deep neural networks in practice are unclear. Thus, the contribution of this paper is limited.

**Summary Of The Review:**

 My main concern is the implications of these results on the modern neural networks are unclear, which weakens the contributions of this paper.

---

> ### Author Response · Authors · 2022-11-15
> **Response to Reviewer oXvD**
>
> We thank the reviewer for their valuable comments. We provide our responses below.
>
> **Relationship with deep neural networks**
>
> Although deep neural networks are the state of the art for many ML tasks, the theoretical understanding of the generalization performance is still far from complete. Specifically, training NNs can be successful without explicit regularizers, and the parametrization of NN induces a non-convex loss. Many papers that investigate these two phenomena theoretically focus on some simplified (and not commonly used) architectures, such as (diagonal) linear NN [1, 2], linear convolutional NN [3], etc. Apart from sparsity and low-rankness, we discover and prove the implicit regularization effect for group-sparsity with a diagonally grouped linear NN. It adds to our understanding of the implicit regularization in training NN. Although there may not be many prior applications for such simplified architectures, we believe the understanding of the related implicit bias will guide the design of new architectures.
>
> **Relationship between the structure of their model and grouped CNNs**
>
> Our design consists of a grouping module and a power module. Our theoretical investigation shows the existence of implicit group sparse regularization when we use a diagonal linear layer for each module. Other types of layers can also be used, for example, CNN in the grouping module and linear (fully connected) layer in the power module. There may exist more interesting implicit regularization effects, which we leave for future work.
>
> **Experimental results to demonstrate the findings exist in real applications**
>
> The major motivation for this work is to understand the implicit regularization effect of gradient descent, and show that group sparsity is one of them. Group structure is useful and common in many problems, such as genomics data [4]. Besides the understanding of training trajectories, our analysis could be useful in architecture design when data is more structured.
>
> Regarding the application in neural networks, we conduct more experiments (see Appendix E.2.). We implement the grouping module (CNN) and power module (FCNN) in an auto-encoder. The implicit regularization effect is visualized as the meaningful interpolation in latent space. As the grouping information is manually added, the effect of grouping CNN is mainly shown in parameter efficiency.
>
>
> [1] Gunasekar, Lee, Soudry, Srebro, Implicit Bias of Gradient Descent on Linear Convolutional Networks, Neurips 2018.
>
> [2] Arora, Cohen, Golowich, Hu, A Convergence Analysis of Gradient Descent For Deep Linear Neural Networks, ICLR 2019.
>
> [3] Nacson, Ravichandran, Srebro, Soudry, Implicit Bias of the Step Size in Linear Diagonal Neural Networks, ICML 2022.
>
> [4] Meier, van de Geer and Buhlmann, The group lasso for logistic regression, JRSSB 2008.

---

> > ### Author Response · Authors · 2022-11-25
> > **More results on real data**
> >
> > To further discover the potential applications of our findings, we conducted more experiments on real data, where group sparsity is anticipated. We use a gene expression dataset from the microarray experiments of mammalian eye tissue samples [2]. The dataset consists of 120 samples with 100 predictors (expanded from 20 genes using 5 basis B-splines, as described in [1]). The goal is to predict the gene expression level of TRIM32, which causes Bardet-Biedl syndrome. We randomly split the data equally, and use the validation dataset for hyperparameter tuning/early stopping. We compare our approach with the commonly used proximal gradient descent and a primal-dual approach. The result is shown below.
> >
> > |Test error|PGD|Primal-Dual|Our approach|
> > |---|:---:|:---:|:---:|
> > |**MSE**|0.03096 |0.02868|0.02477|
> >
> > We look forward to your reply.
> >
> > [1] Yang and Zhou, A Fast Unified Algorithm for Computing Group-Lasso Penalized Learning Problems, Statistics and Computing, 2015.
> >
> > [2] Scheetz et al, Regulation of gene expression in the mammalian eye and its relevance to eye disease, PNAS, 2006.

---

### Decision · Program_Chairs · 2023-01-20

**Decision:**

Accept: poster

**Justification For Why Not Higher Score:**

Comparison with explicit regularization methods could be reinforced.
Similarly, the practical implications are not direct for standard NN architectures.

**Justification For Why Not Lower Score:**

It is a well-organized and well written paper with potentially impactful results.


**Metareview: Summary, Strengths And Weaknesses:**

The paper studies the implicit bias of gradient descent on diagonal linear networks.
A special architecture is proposed so gradient descent imposes group sparsity on the network weights.
An analysis is performed under the (strong) assumption that the features are orthogonal.
Gradient flow of the overparametrized objective is analyzed, and then a practical version of the gradient flow is introduced (gradient descent with weight normalization).
Guaranties similar to the sparse recovery literature are derived under sub-Gaussianity of the noise.

**Note From Pc:**

if the above contains the word "oral" or "spotlight" please see: "oral" presentation means -> notable-top-5% and "spotlight" means -> notable-top-25%. As stated in our emails, we are disassociating presentation type from AC recommendations